# Enhanced carbon dioxide electrolysis at redox manipulated interfaces

Wenyuan Wang[1], Lizhen Gan[1,2], John P. Lemmon[3], Fanglin Chen [4], John T.S. Irvine[1,5] & Kui Xie [1]

Utilization of carbon dioxide from industrial waste streams offers significant reductions in global carbon dioxide emissions. Solid oxide electrolysis is a highly efficient, high temperature approach that reduces polarization losses and best utilizes process heat; however, the technology is relatively unrefined for currently carbon dioxide electrolysis. In most electrochemical systems, the interface between active components are usually of great importance in determining the performance and lifetime of any energy materials application. Here we report a generic approach of interface engineering to achieve active interfaces at nanoscale by a synergistic control of materials functions and interface architectures. We show that the redox-manipulated interfaces facilitate the atomic oxygen transfer from adsorbed carbon dioxide molecules to the cathode lattice that determines carbon dioxide electrolysis at elevated temperatures. The composite cathodes with in situ grown interfaces demonstrate significantly enhanced carbon dioxide electrolysis and improved durability.

---

[1] Key Laboratory of Design and Assembly of Functional Nanostructures, Fujian Institute of Research on the Structure of Matter, Chinese Academy of Sciences, 350002 Fuzhou, Fujian, China. [2] School of Transportation and Civil Engineering, Fujian Agriculture and Forestry University, No.15 Shangxiadian Road, 350002 Fuzhou, Fujian, China. [3] National Institute of Clean and-Low-Carbon Energy (NICE), 102211 Beijing, China. [4] Department of Mechanical Engineering, University of South Carolina, 300 Main Street, Columbia, SC 29208, USA. [5] School of Chemistry, University of St Andrews, St Andrews, Fife KY16 9ST, UK. Correspondence and requests for materials should be addressed to J.T.S.I. (email: jtsi@st-andrews.ac.uk) or to K.X. (email: kxie@fjirsm.ac.cn)

Sustainable future energy scenarios require reliable large-scale electricity storage/utilization in order to address the intermittent nature of renewable power sources. Electrochemical energy conversion and storage technologies are promising routes while in most electrochemical systems the interface between active components is usually of great importance in determining functionality in any energy materials application. The critical region determining the performance and lifetime of most electrochemical devices is normally at the electrode side of the electrode/electrolyte interface. A generic approach of interface engineering on the nanoscale by a synergistic control of materials functions and interface architectures would be disruptive to enhance both electrode performance and durability. Solid oxide electrolysis of $CO_2$ is expected to play a pivotal role in the transition to a carbon neutral energy landscape based on renewable sources[1,2]. The critical electrode active region is located within a zone only several micrometers from electrolyte surface. These active areas are often at the so-called three-phase boundary (TPB) with the convergence of electronic conduction phase, ionic conduction phase, and gaseous phase[3].

There are three major SOE cathode types including metal, metal/oxide cermet, and ceramic cathodes[4]. The active interface for metallic cathodes where $CO_2$ splitting reactions proceed is normally located on the electrolyte surface with intimate contact with porous metal phase. Although metallic cathodes have demonstrated promising performances[5], there are still critical issues related to passivation at these interfaces even under the requisite reducing atmosphere. In comparison, the state-of-the-art Ni–YSZ (YSZ, yttria stabilized zirconia) composite cathode with mixed conductivity is preferred. The active interfaces, TPBs, are located at the exposed surface of the contact point of Ni and YSZ in the cathode and their fine structures dominate both cathode activity and durability[6,7]. Similar to metal cathodes, $CO_2$ electrolysis at the Ni–YSZ composite cathode needs to be operated in a reducing atmosphere, such as flowing CO or $H_2$ to avoid oxidation of Ni phase[8]. In contrast, a ceramic cathode with mixed conduction has a more extensive surface, functioning as electrochemical reaction region. Nevertheless, low performance is often encountered with ceramic cathode mainly due to low catalytic activity though the redox stability brings durability advantages.

For a given cathode, the overall interface architectures determine cathode performance and durability. Active interfaces have been constructed through loading nanoparticles particularly using an impregnation method for all these three major types of cathodes for $CO_2$ electrolysis[9]. For example, loading of metal or oxide nanoparticles in porous nickel, Ni–YSZ scaffold and titanate electrode to ex situ assemble interfaces, all have demonstrated the successful manipulation of nanostructures at interfaces aiming at electrode activity enhancement. Although overall cell performance including coking, poisoning, and oxidation resistances have been improved under some operation conditions, long-term stability has been shown to be limited by the nanoparticle agglomeration with diminishing active interfaces. In contrast, we have recently demonstrated the reversible in situ exsolution of metal nanoparticles on perovskite oxide scaffolds through a phase decomposition process during operation[10]. In situ growth of uniformly dispersed metal nanoparticles can be triggered under reducing conditions or applied potentials. The Ni-anchored titanates not only favorably show high performance but also demonstrate excellent stability[11]. Consequently, interface engineering at nanoscale can control the number of available electroactive sites and governs the cathode and cell performances while the interface structures are the key factors that determine the durability in operation[3,12].

The electrochemical $CO_2$ splitting in cathode can be considered as a defect reaction that represents a Kroger–Vink notation

($CO_2 + V_o^{..} + 2e \rightarrow CO + O_o^x$) involving a key step of atomic oxygen transfer, namely, the transfer of oxygen atoms from the adsorbed $CO_2$ to occupy the oxygen vacancy in the cathode lattice. The complicated electrode process includes $CO_2$ molecular adsorption, activation, splitting, and atomic oxygen diffusion to the oxide lattice; however, the rate-limiting step is normally considered to be the atomic oxygen diffusion from the adsorbed $CO_2$ into the bulk of the electrode[13–15]. The atomic oxygen transfer from adsorbed $CO_2$ to oxygen vacancy in the oxide lattice is highly limited due to the slow oxygen exchange rates between two different phases in contrast to the efficient molecular $CO_2$ adsorption and activation[16,17]. The electrocatalysis activity mainly originates from the interfacial gas–solid reaction at active interface region while the oxygen transfer process determines $CO_2$ splitting kinetics that accordingly governs the cathode activity. In situ exsolved metal/oxide interfaces with strong interactions would provide the possibility to facilitate atomic oxygen transfer process. These exsolved metal–oxide interface through phase decomposition has been demonstrated to be effective to enhance $CO_2$ electrolysis.

In this work, we develop new generic approaches of manipulation of active metal–oxide interfaces considering cathodes ranging from metallic nickel to Ni–YSZ cermet, $CeO_{2-\delta}$, and $Nb_{1.33}(Ti_{0.8}M_{0.2})_{0.67}O_4$ (M = Mn,Cr) ceramic compositions, through control of phase decomposition during reduction. These in situ grown metal or oxide nanoparticles on porous cathodes produce active metal–oxide interface that would function as TPB at nanoscale. The anchored nanoparticles confined on porous scaffolds not only dramatically promote cathode performance but also enhance durability. We investigate the oxygen transfer rates of cathode with exsolved interfaces and then study $CO_2$ electrolysis.

## Results

**Interface growth.** We firstly demonstrate the in situ exsolution of interfaces with nanosized $MnO_x$ anchoring on metallic nickel cathode using a phase decomposition process. The growth of oxide islands on metallic nickel scaffold generates active metal–oxide interfaces. Figure 1a shows the X-ray diffraction (XRD) of mixtures of cubic NiO and spinel-type $NiMn_2O_4$ (0–20 wt%) that are used to transform into metallic nickel cathode after reduction. Figure 1b reveals the presence of a new cubic $MnO_x$ phase upon reduction. In this case, NiO is reduced to Ni, while spinel $NiMn_2O_4$ decomposes into metallic Ni and $MnO_x$. As expected, nickel element is exclusively present as $Ni^{2+}$ while manganese is present in the form of $Mn^{2+}$, $Mn^{3+}$, and $Mn^{4+}$ in the sample, as shown in X-ray photoelectron spectroscopy (XPS) in Supplementary Fig. 1[18]. Upon reduction, only metallic nickel is observed while manganese is predominantly reduced into $Mn^{2+}$, with some $Mn^{3+}$ still present. This corroborates with the XRD results, suggesting that metallic nickel and $MnO_x$ are the main phases after reduction. The oxygen nonstoichiometry of reduced $MnO_x$, in Supplementary Fig. 2, is determined to be $MnO_{1.10}$ using thermogravimetric analysis (TGA) analysis, which further confirms the dominant presence of $Mn^{2+}$. Similarly, the exsolution of $MnO_x$ has been achieved in a Ni–YSZ cermet as shown in Supplementary Fig. 3 when using $NiO/NiMn_2O_4$ precursor combined with YSZ to construct a $MnO_x$-anchored composite cathodes.

We then consider in situ exsolved active interfaces on an oxide scaffold that involves metal nanoparticles growing and anchoring on oxide substrate with linked creation of oxygen vacancies. In Fig. 1c, d, the Ni/Cu is doped into the ceria lattice during synthesis while the $Ni_{1-x}Cu_x$ nanoparticles are exsolved to anchor on ceria surface to construct the $Ni_{1-x}Cu_x$–ceria interface.

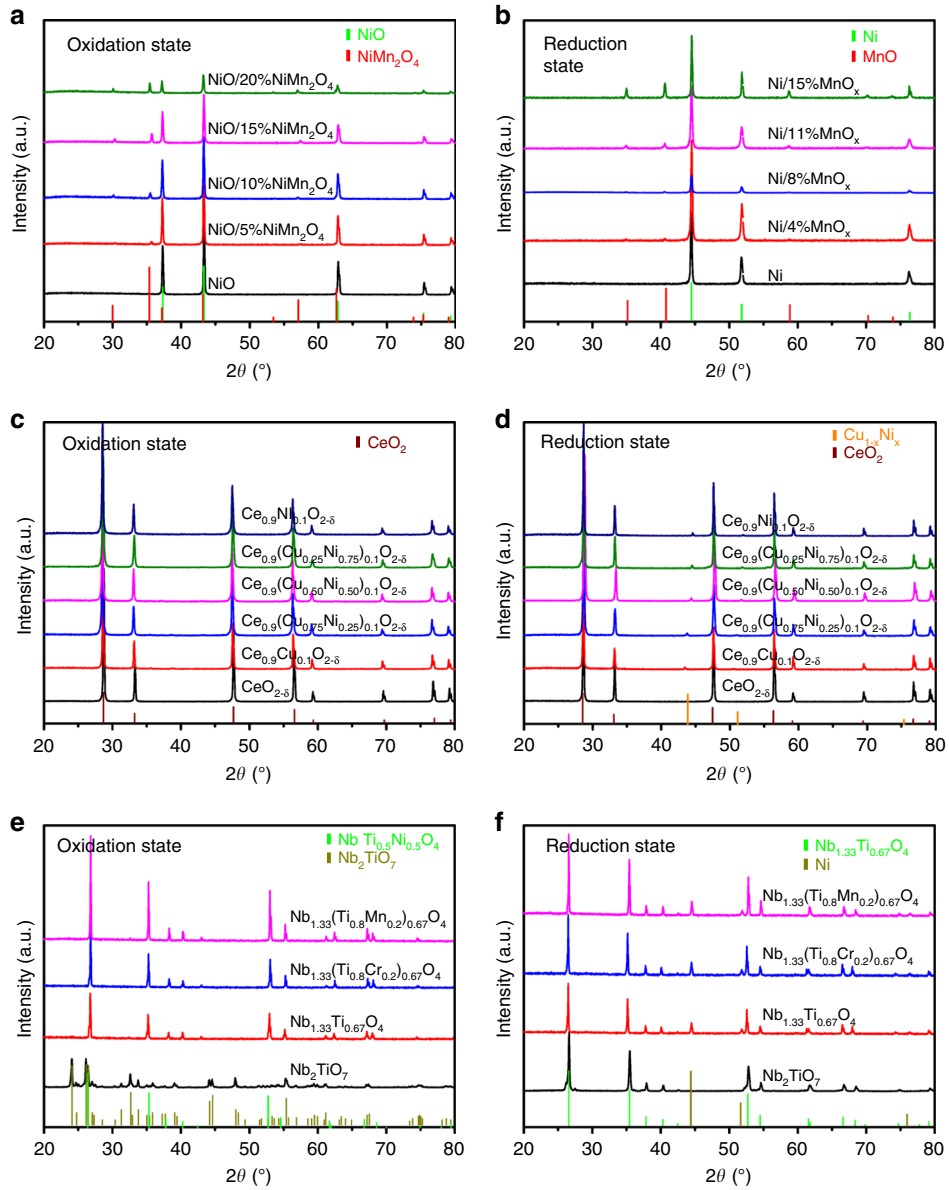

**Fig. 1** X-ray diffraction patterns of samples. **a** Oxidized samples of NiO/$x$%NiMn$_2$O$_4$ ($x$ = 0, 5, 10, 15, 20). **b** Reduced samples of Ni/$y$%MnO$_x$ ($y$ = 0, 4, 8, 11, 15). **c** Oxidized and **d** reduced samples CeO$_2$ and Ce$_{0.9}$(Cu$_{1-x}$Ni$_x$)$_{0.1}$O$_{2-\delta}$ ($x$ = 0, 0.25, 0.5, 0.75, 1). **e** Oxidized and **f** reduced samples Nb$_2$TiO$_7$, NbTi$_{0.5}$Ni$_{0.5}$O$_4$, NbTi$_{0.4}$Cr$_{0.1}$Ni$_{0.5}$O$_{3.95}$, and NbTi$_{0.4}$Mn$_{0.1}$Ni$_{0.5}$O$_4$

After reduction, up to ~90% of the metal is exsolved from lattice according to the TGA analysis in Supplementary Fig. 2, while the generation of oxygen vacancies (Ce$^{4+}$ → Ce$^{3+}$) would strongly couple with metal nanoparticles[19]. The active interface can be reversibly integrated back into backbone when exposed in oxidizing atmospheres at up to 600 °C as shown in the in situ XRD in Supplementary Fig. 3. XPS confirms that the Ni/Cu elements are present at +2 oxidation state while the Ce$^{4+}$ is dominant in the oxidized samples. After reduction, only Ni$_{1-x}$Cu$_x$ alloys are present while part of the Ce$^{4+}$ has been reduced into Ce$^{3+}$ as shown in Supplementary Fig. 4[20].

Similarly, we show in situ exsolved interface on (Nb,Ti)O$_4$ electronic conductor while ionic conductivity can be enhanced through the doping of Mn/Cr to create oxygen vacancies. During synthesis, the Ni is doped in the lattice while it is then exsolved after reduction. Figures 1e, f show the XRD patterns of the Nb$_{1.33}$Ti$_{0.67}$O$_4$, NbTi$_{0.5}$Ni$_{0.5}$O$_4$, Nb$_{1.33}$(Ti$_{0.8}$Mn$_{0.2}$)$_{0.67}$O$_4$, and Nb$_{1.33}$(Ti$_{0.8}$Cr$_{0.2}$)$_{0.67}$O$_4$ samples before and after reduction. The

Ni-containing sample is in rutile structure whereas the parent is a layered defect variant. In Fig. 1e, the substitution of Ti by Ni, Mn, or Cr in single-phase NTO confirms the homogeneous solid solution with different Cr/Mn/Ni dopants[21]. As demonstrated in Fig. 1f, the phase transformation is observed for all samples after reduction while Ni metal is grown on Nb$_{1.33}$Ti$_{0.67}$O$_4$ and Nb$_{1.33}$(Ti$_{0.8}$M$_{0.2}$)$_{0.67}$O$_4$ (M = Mn, Cr) in which the oxygen vacancies are mainly linked to Mn/Cr dopants. The valence change of Ni, Mn, and Ti in XPS in Supplementary Fig. 5 further confirms the phase changes before and after reduction. In this case, up to ~90% of Ni is exsolved according to TGA analysis in Supplementary Fig. 2, while the oxygen vacancy concentrations are 0.1 mol for the reduced Mn-doped and Cr-doped samples. These oxide scaffold with mixed conducting property[22] would favor the activity enhancement of the metal–oxide interfaces in cathode.

Microstructural investigations are performed to study the interface architectures on different reduced cathodes. In situ

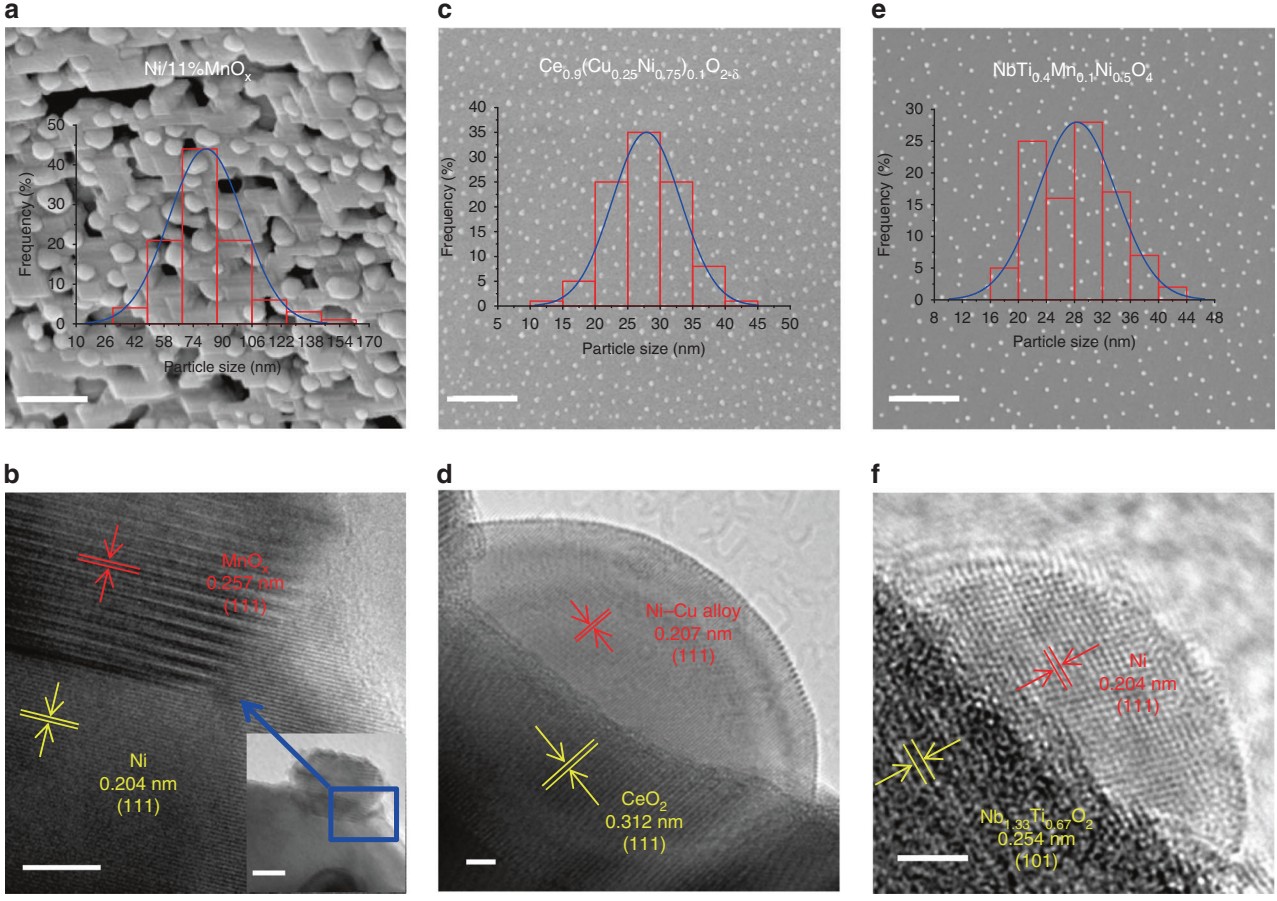

**Fig. 2** SEM and HRTEM micrographs for samples. **a**, **b** Reduced Ni/11%MnO$_x$ samples. **c**, **d** Reduced Ce$_{0.9}$(Cu$_{0.25}$Ni$_{0.75}$)$_{0.1}$O$_{2-\delta}$ sample. **e**, **f** Reduced NbTi$_{0.4}$Mn$_{0.1}$Ni$_{0.5}$O$_4$ samples. The scale bar is 5 nm in **a–c** and **e**. The scale bar is 5 nm in **b** and the scale bar is 10 nm in the inset in **b**. The scale bar is 2 nm in **d** and **f**

grown interface architectures not only avoid any possible agglomeration of nanoparticles at high temperatures, that would occur for infiltrated particles, but also create of catalytically active sites. In Fig. 2a, the MnO$_x$ nanoislands have formed on a porous nickel scaffold, indicating that catalytically active particles can be in situ grown through reduction. The population density of MnO$_x$ nanoparticles (~20 nm) significantly increases with MnO$_x$ weight percentage varying from 8% to 15% in Fig. 2a and Supplementary Fig. 6. In Fig. 2b, high-resolution transmission electronic microscopy (HRTEM) image of Ni/MnO$_x$ sample shows a clear heterojunction between the two phases, with the MnO$_x$ nanoparticle deeply embedded into the nickel surface. The good anchoring of MnO$_x$ particles onto the Ni support suggests not only a potentially catalytically active site, but also a strong interfacial bond, which would benefit long-term thermal and chemical stability. In contrast, the in situ exsolution of active interfaces on ceria cathode can be achieved through the reversible growth of metal nanoparticles. In Fig. 2c, the Ni$_{0.5}$Cu$_{0.5}$ alloy (~50 nm) uniformly distributes on ceria surface while the metal–ceria heterojunction interfaces in Fig. 2d are expected to deliver strong interactions. Similarly, active interface structures are exsolved on rutile Nb$_{1.33}$(Ti$_{0.8}$M$_{0.2}$)$_{0.67}$O$_4$ (M = Mn, Cr) in which the doping of Mn/Cr creates oxygen vacancies. Notably, Ni nanoparticles strongly anchor onto the Nb$_{1.33}$(Ti$_{0.8}$Mn$_{0.2}$)$_{0.67}$O$_4$ surface after the reduction in Fig. 2e, while the metal–oxide interfaces are shown in Supplementary Fig. 6. The TEM results in Fig. 2f further demonstrate the reversible exsolution of anchored interfaces with heterojunction contacts between metal and oxide.

**Interface activity**. Figure 3a shows the normalized conductivity profiles collected at 800 °C with the atmospheres alternating between two oxygen partial pressures ($1 \times 10^{-18} \to 1 \times 10^{-12}$ atm, CO/CO$_2$ mixtures). For the ceria system, the rebalance time of conductivity remarkably reduces from 6400 to 600 s for the samples with in situ growth of exsolved interfaces, while the oxygen exchange coefficient ($K_{ex}$) is enhanced by ~15 times from $2.0 \times 10^{-5}$ to $2.92 \times 10^{-4}$ cm s$^{-1}$. In this process, the oxygen transfer from adsorbed CO$_2$ to oxide lattice would reach an equilibrium, and the oxygen exchange coefficient is indeed an oxygen transfer coefficient. Meanwhile, the rebalance time of conductivity remarkably reduces from 15490 to 535 s for titania system with in situ growth of exsolved metal nanoparticles, while the oxygen exchange coefficient ($K_{ex}$) is enhanced by ~7 times from $2.6 \times 10^{-5}$ to $1.78 \times 10^{-4}$ cm s$^{-1}$ (Fig. 3b). The exsolved nanoparticles would create active metal–oxide interfaces that may facilitate the oxygen transfer process. Here we summarize the relationship between oxygen exchange coefficient ($K_{ex}$) and the electrochemical performances in Fig. 3c, d. The alloy nanoparticles with intimate interaction between different metals may deliver excellent performance for oxygen transfer at interfaces. For ceria system, the Ni$_{1-x}$Cu$_x$ alloy nanoparticles at interfaces deliver remarkable advantage for oxygen transfer process. Accordingly, the electrode polarization resistance and electrochemical process are significantly improved. The electrochemical performance of reduced NbTi$_{0.4}$Mn$_{0.1}$Ni$_{0.5}$O$_4$ enhances by ~100% compared with reduced NbTi$_2$O$_7$. The coupling of in situ exsolved interface and oxygen vacancies created by Mn/Cr

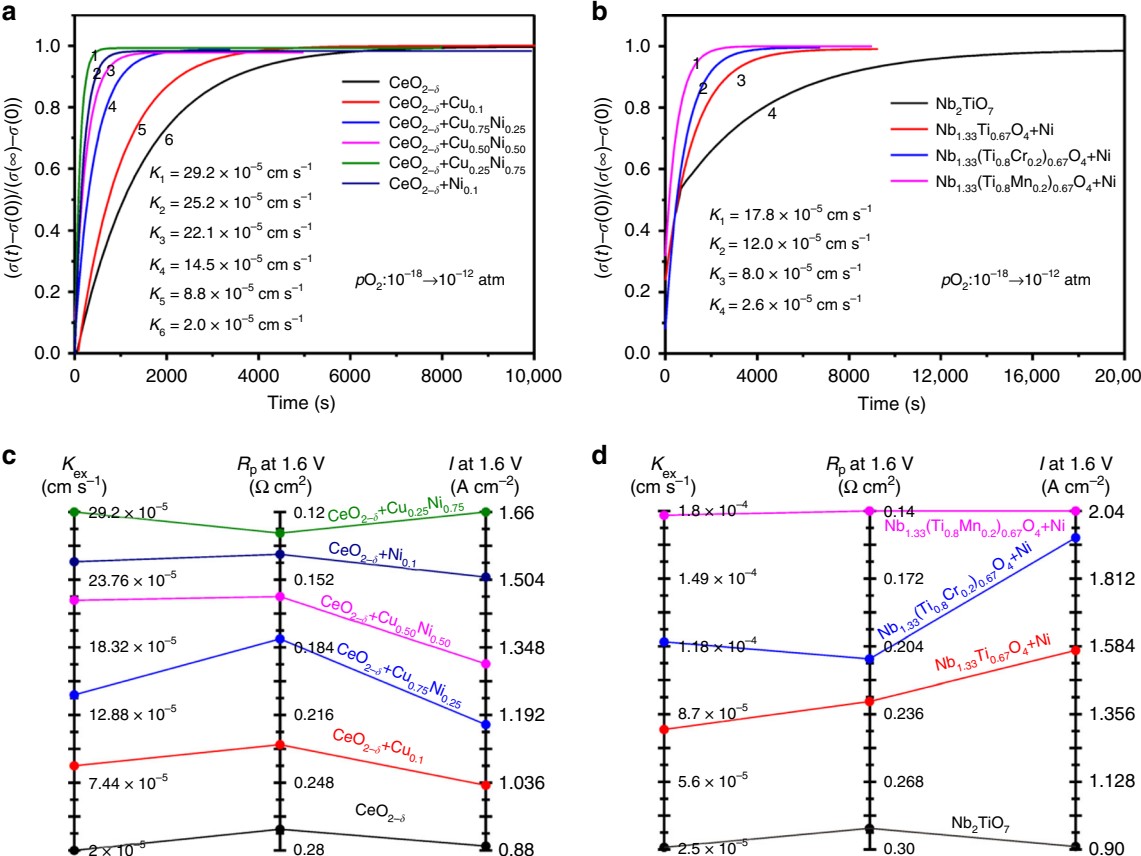

**Fig. 3** Oxygen transfer at interfaces tested using electrical conductivity relaxation. **a, b** Normalized conductivity profiles of reduced $Ce_{0.9}(Cu_{1-x}Ni_x)_{0.1}O_{2-\delta}$ ($x = 0$–0.1) and $TiO_2$ system ($Nb_2TiO_7$, $NbTi_{0.5}Ni_{0.5}O_4$, $NbTi_{0.4}Cr_{0.1}Ni_{0.5}O_4$, and $NbTi_{0.4}Mn_{0.1}Ni_{0.5}O_4$). **c, d** The relationship between surface exchange coefficient ($K_{ex}$) and polarization resistance or current density at 1.6 V

dopant improves oxygen transfer ability that hence enhance electrode activity for $CO_2$ electrolysis.

In situ fourier transforming infrared spectrum (FT-IR) spectroscopy at 800 °C is utilized to probe $CO_2$ on reduced cathode surfaces. In Fig. 4a, b, the Ni/y%MnO$_x$ ($y = 4, 8, 11, 15$) shows absorption in two distinct IR bands, i.e. 2400–2300 cm$^{-1}$, which is typically observed for molecular $CO_2$[23], and 1400–1350 cm$^{-1}$, which is usually associated with carbonate ions ($CO_3^{2-}$)[24], suggesting that the adsorbed $CO_2$ is present in an intermediate state between the molecular and ionic species. In contrast, no $CO_2/CO_3^{2-}$ absorption can be detected on the bare nickel, suggesting that the presence of $MnO_x$ nanoparticles significantly enhances $CO_2$ chemisorption. On the contrary, oxide cathodes with oxygen nonstoichiometry are more favorable for $CO_2$ accommodation on surfaces. In Fig. 4c–f, similar $CO_2$ adsorption phenomenon are observed for reduced samples based on ceria and $(Nb,Ti)O_4$ up to high temperatures, while the metal itself contributes little to chemisorption. We further calculate the FT-IR spectrum of chemisorption of $CO_2$ on these oxide surfaces as the dominant surface would give general FT-IR signals in experiments. To simplify the calculation, we calculate the vibrational frequencies of $CO_2$ species and $CO_3^{2-}$ species adsorbed on MnO, $CeO_2$, and $TiO_2$ surfaces. We calculate the vibrational frequency assuming a stable surface facet of oxide to simplify the calculation model, though the oxide surface is extremely complicated and is composed of poly-crystalline or amorphous states. As shown in Supplementary Fig. 7 and Supplementary Table 1, the vibrational frequencies of $CO_2$ are about 2370 cm$^{-1}$, which is mainly from physical adsorption on

the surfaces. And the vibrational frequencies of carbonate ($CO_3^{2-}$) are fitting the experimental data well. And the presence of oxygen vacancies linked to dopants on oxide surfaces are highly favorable to $CO_2$ reduction in a exothermic process[25].

These critical regions at exsolved interfaces would determine the performance and lifetime of solid oxide cell; however, it is very hard to in situ study the $CO_2$ splitting at these active metal–oxide interfaces. Theoretical calculations are performed to provide mechanistic insights into the $CO_2$ adsorption/activation on nanoparticle, substrate, and metal–oxide interfaces. The different adsorption configurations of the models of cluster structure on substrate are considered and shown in Supplementary Fig. 8a after the optimization. For $MnO_x/Ni(111)$ system, the possible $CO_2$ adsorption configurations are shown in Supplementary Fig. 9a, where $CO_2$ forms a bidentate with higher adsorption energy when $MnO_x$ is highly reduced. The C atom of $CO_2$ binds with Ni, whereas at least one O atom binds with Mn or both Mn and Ni. A short C–Ni bond length of 1.87–1.91 Å indicates a strong interaction between the nickel surface and $CO_2$. The Mn–O and Ni–O bonds that are formed upon adsorption are also shorter than that observed in typical Mn–O (1.80–2.70 Å) and Ni–O (2.05–2.08 Å)[26,27], which is indicative of a strong chemical interaction. The elongated C–O bonds and bent O–C–O angles in $CO_2$ additionally suggest significant activation upon adsorption. An overview of adsorption energies, bond distances, and angles are provided in Supplementary Table 2. Figures 5a, b shows the changes in electronic charge density that takes place upon $CO_2$ chemisorption. Both Mn and Ni donate electron density mainly to C and the O atoms of $CO_2$. These changes take

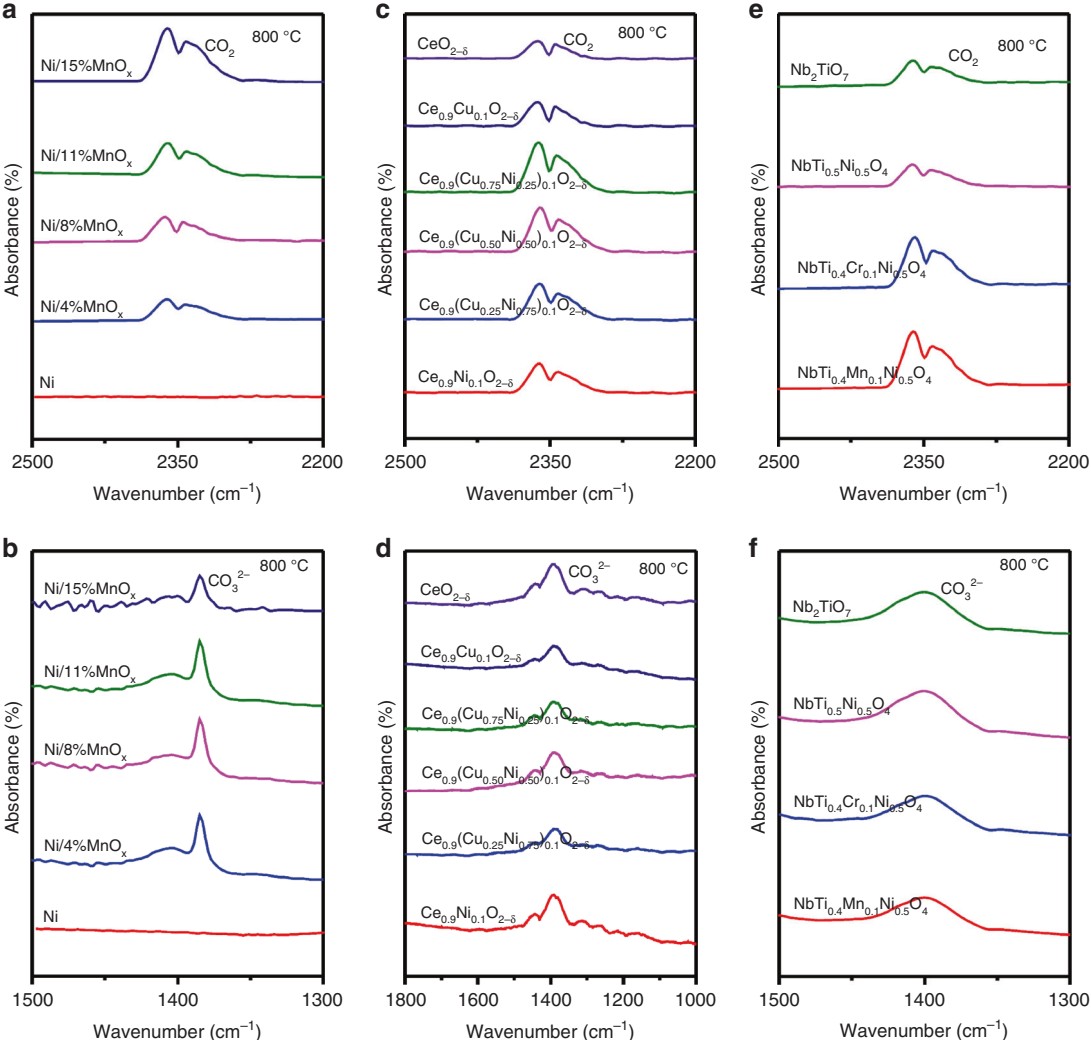

**Fig. 4** Chemical adsorption of $CO_2$ for different samples. **a**, **c**, and **e** In situ FT-IR spectroscopy of $CO_2$ molecule absorbed on different reduced samples at 800 °C. **b**, **d** and **f** In situ FT-IR spectroscopy of carbonate ion absorbed on different reduced samples at 800 °C

place mainly within the $2p$ orbitals of the involved atoms. The increase of electron density of C, O1, and O2 results in the elongation of $CO_2$, thus promoting the activation of adsorbed $CO_2$ molecule.

For the $Ni_{1-x}Cu_x/CeO_{2-\delta}$ system, the clusters of Cu, Ni, and Ni–Cu on the $CeO_2(111)$ surface are considered for $CO_2$ chemisorption. Figure 5c shows the most stable adsorption configurations while other possible states are shown in Supplementary Fig. 9b–d. The Ni–Cu alloy cluster gives the most stable chemisorption configuration ($-2.06$ eV) at interfaces in contrast to $-1.32$ eV for Cu and $-1.73$ eV for Ni at metal–oxide interfaces, and all these adsorption energies are larger than that of chemisorption on $CeO_2$ surfaces. For the stable chemisorption at interfaces, the C atoms of $CO_2$ bind with surface O atoms, whereas at least one O atom binds with Ni or both Ce and Ni. The Ni–O and Ce–O distances are 1.86–2.00 and 2.44–2.55 Å, respectively. More importantly, the oxygen vacancy at interface further enhances $CO_2$ activation in Fig. 5d, and the highest adsorption energy ($-2.18$ eV) is observed for the defected $(Ni–Cu)/CeO_2$ system. The $O_2$ of the $CO_2$ molecules embed into the surface defects while C atoms bind with Ni atoms of Ni–Cu clusters with the bond length of 1.85 Å for C–Ni. The adsorption energies, bond distances, and angles are provided in

Supplementary Table 3. The charge density of $CO_2$ adsorption on clean and defect site mainly changes in $2p$ orbitals of the C atoms and $O_2$ atoms of $CO_2$, Ni atoms, and the relevant surface Ce atoms. The adsorbed $CO_2$ gains some electron density donated by the surrounding Ce/Ni/Cu atoms, indicating the effective charge transfer from the metal–oxide system to $CO_2$ molecules. Similar phenomenon has also been observed for $CO_2$ chemisorption at defected $Ni/(Nb,Ti)O_4$ system in Fig. 5e, f, Supplementary Fig. 9e, and Supplementary Table 4. In summary, the metal–oxide interface is favorable for $CO_2$ chemisorption and activation.

**Carbon dioxide electrolysis.** Electrolysis of $CO_2$ is performed for each of these variants of nanoparticles-decorated cathodes. The metallic nickel cathodes are investigated with $80\%CO_2/2\%CO/Ar$ under applied voltages of 0.4–1.6 V at 800 °C. The current–voltage ($I$–$V$) curves clearly reveal the superior performance with 300% enhancement for the porous nickel with in situ growth of $MnO_x$ nanoparticles in comparison to the bare nickel cathode (Supplementary Fig. 10). The exsolution of $MnO_x$ nanoparticles drastically improves the current density to 3.1 A cm$^{-2}$ at 1.6 V when the optimum $MnO_x$ nanoparticle contents are obtained as summarized in Fig. 6a. These values are much higher than the reported performance with metallic nickel

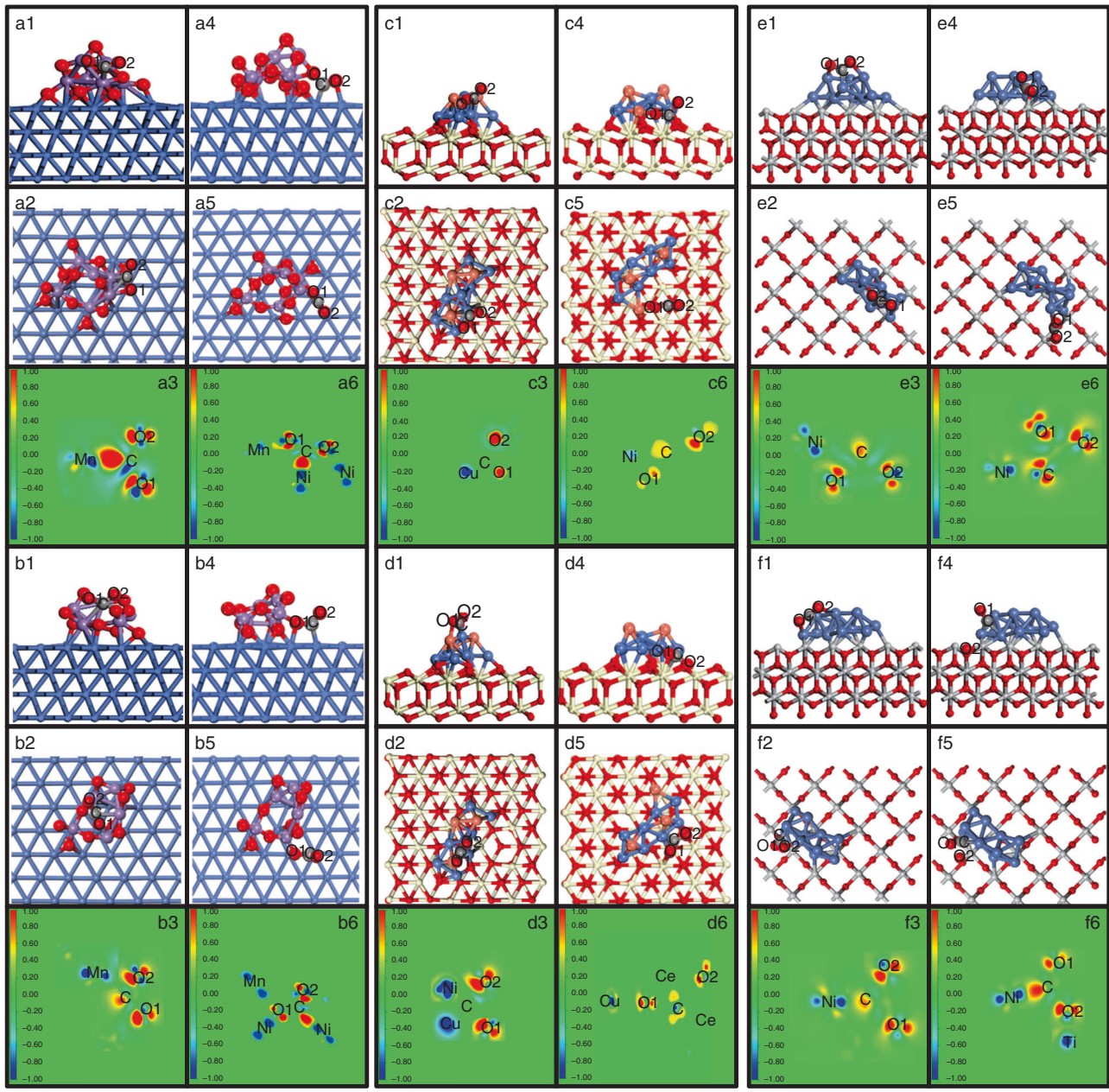

**Fig. 5** Different adsorption configurations of $CO_2$ on the system. **a**, **b** The $MnO_x$/Ni system. **c**, **d** The Ni–Cu/$CeO_2$ system. **e**, **f** The Ni/$TiO_2$ system (top: clean surfaces, bottom: defective surfaces, 1 and 4: side views, 2 and 5: top views, 3 and 6: contour plots of electronic charge density difference). Nickel in blue, copper in orange, cerium in silvery white, titanium in pale, manganese in purple, oxygen in red, and carbon in gray

even decorated with $LaFeO_3$ oxide[28]. Similarly, the exsolution of Ni/$MnO_x$ interfaces significantly improves the performance of Ni–YSZ cathode by 300% under identical conditions with the optimum $MnO_x$ nanoparticles of 3% in weight ratio. The exsolved interfaces remarkably improve ceria cathode performance while the $Cu_{1-x}Ni_x$ alloy effect delivers the best performance for $Cu_{0.25}Ni_{0.75}$–ceria cathode as summarized in Fig. 6a. The observed current density with $Cu_{0.5}Ni_{0.5}$–ceria cathode reaches ~1.3 A $cm^{-2}$ at 1.6 V which is 200% enhancement in contrast to bare ceria cathode. For $(Nb,Ti)O_4$ cathodes, the current density is enhanced through the exsolution of Ni nanoparticles while significant improvement is achieved when oxygen vacancies linked to Cr/Mn dopants are created to facilitate metal–oxide interface interactions. The current densities with Ni/$Nb_{1.33}(Ti_{0.8}M_{0.2})_{0.67}O_4$

($M = Mn,Cr$) cathodes reach ~1.6 A $cm^{-2}$ with 200% enhancement at 1.6 V and 800 °C.

As summarized in Fig. 6b, the exsolved interfaces strongly improve electrode polarization resistances. The AC impedance under an operation condition is shown in Supplementary Figs. 11–14 and modeled using Zview software. As summarized in Fig. 6c, d, the cathodes with optimum exsolved interfaces show the exceptionally high CO generation rates and Faradaic efficiencies. Although the applied voltages effectively tailor electrode activity and CO production, the best performance is observed for $Cu_{0.5}Ni_{0.5}/CeO_{2-\delta}$ and Ni/$Nb_{1.33}(Ti_{0.8}M_{0.2})_{0.67}O_4$ ($M = Mn,Cr$) cathodes. In this case, the growth of exsolved interfaces at nanoscale delivers enhanced electrocatalytic activity. We further study the durability of direct $CO_2$ electrolysis at

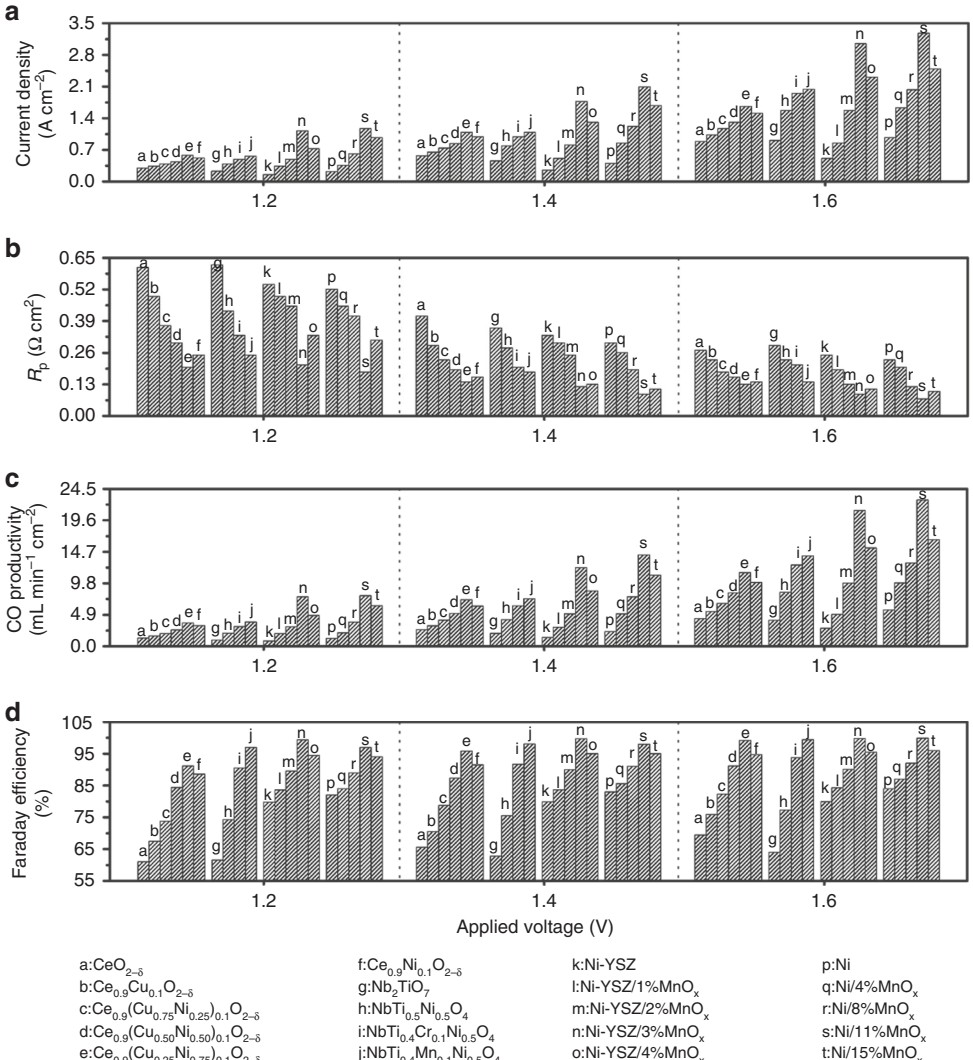

**Fig. 6** Electrochemical performance tested at 800 °C. **a** Current density of $CO_2$ electrolysis with different cathodes. **b** Polarization resistance of cells with different cathodes. **c**, **d** CO productivity and Faraday efficiency with different cathodes

800 °C using metallic and ceramic cathodes in Supplementary Fig. 15. The metallic cathode with exsolved nanoparticles has demonstrated a stable performance in pure $CO_2$ at 1.4 V for up to 100 h, which confirms the substantially enhanced oxidation resistance of metallic nickel cathode. The ceria cathode shows a stable current density during the electrolysis operation at 0.58 mA cm$^{-2}$, which further confirms the stability of exsolved interfaces.

## Discussion

In conclusion, enhanced $CO_2$ electrolysis has been achieved with an interface engineering in a wide range of cathodes from metal to ceramic compositions through control of phase decompositions during reduction. These nanoscale-active architectures not only involve strongly anchored functional phase at the interfaces but also consist of strong coupling of oxygen vacancies with metal catalyst. The in situ exsolution of metal/oxide interfaces at nanoscale produce strong interaction that significantly enhances oxygen transfer at interfaces. These interface architectures with anchored and confined nanosized particles not only dramatically promote cathode performance but also enhance durability. This work provides a general design guidance in SOE cathodes for high performance $CO_2$ electrolysis and would shed light on the

interface tailoring between active components in determining the performance and lifetime of energy materials application in other electrochemical systems.

## Methods

**Synthesis**. The NiMn$_2$O$_4$ powders are synthesized through a combustion method and calcined at 1200 °C for 6 h in air[29]. For Ni/MnO$_x$ cathode, NiO, and NiMn$_2$O$_4$ are mixed with 0–20 wt% weight ratio of NiMn$_2$O$_4$. For Ni–YSZ/MnO$_x$ cathode, NiO:YSZ (50:50) and NiMn$_2$O$_4$ powders are homogeneously mixed with 0–6 wt% weight ratio of NiMn$_2$O$_4$. The Ce$_{0.9}$(Cu$_{1-x}$Ni$_x$)$_{0.1}$O$_{2-\delta}$ ($x$ = 0, 0.25, 0.5, 0.75, 1) are synthesized by a combustion method. Nb$_{1.33}$Ti$_{0.67}$O$_4$, NbTi$_{0.5}$Ni$_{0.5}$O$_4$, NbTi$_{0.4}$Mn$_{0.1}$Ni$_{0.5}$O$_4$, and NbTi$_{0.4}$Cr$_{0.1}$Ni$_{0.5}$O$_{3.95}$ powders are prepared using a solid-state reaction method. The Ba$_{0.5}$Sr$_{0.5}$Co$_{0.8}$Fe$_{0.2}$O$_{3-\delta}$ (BSCF) powders are synthesized using a sol–gel method and calcined at 1000 °C for 5 h in air. The Ce$_{0.8}$Sm$_{0.2}$O$_{2-\delta}$ (SDC) powders are synthesized using a combustion method[30]. La$_{0.9}$Sr$_{0.1}$Ga$_{0.8}$Mg$_{0.2}$O$_{3-\delta}$ (LSGM) electrolyte is prepared by a conventional solid state reaction method[28].

**Characterization**. The phase formations of the samples are analyzed using XRD (Miniflex600, Rigaku). HRTEM (TECNAI F20, FEI) is utilized to examine interface exsolution. XPS (ESCALAB 250Xi, Thermofisher) is used to analyze the chemical states of the samples before and after reduction. TGA test (TGA, STA449F3, NETZSCH) is used to determine the oxygen nonstoichiometry. The sample microstructures are investigated by scanning electron microscopy (SEM, SU-8010, Hitachi). The reduction of samples was performed in a reducing atmosphere (5% H$_2$/Ar, $p$O$_2$ at ~10$^{-18}$ atm). Interfacial oxygen transfer rates of the sintered samples

with metal–oxide interfaces are tested using an electrical conductivity relaxation (ECR) method[31].

**Electrochemical test**. The LSGM electrolyte support is mechanically polished and then used to assemble cells with different cathodes and BSCF anode. The single cells are sealed using a ceramic paste (JD-767, Jiudian, Dongguan, China) for electrochemical measurements. The cathode is pre-reduced at 800 °C by flowing 5%H$_2$/Ar. $CO_2$ electrolysis is subsequently performed in $CO_2$ (ceramic cathode) or 80%$CO_2$/2% CO/18%Ar (Ni and Ni–YSZ). The gas flow rate is 50 mL min$^{-1}$, whereas the anode is exposed to ambient air. The current–voltage curves (I–V, 0.007 V s$^{-1}$) and in situ AC impedance spectra (4 MHz–0.1 Hz, 10 mV perturbation voltage) are recorded using an electrochemical station (IM6, Zahner). The production of CO is analyzed using an online gas chromatograph (GC2014, Shimazu, Japan).

**Theoretical calculation**. Density functional theory (DFT) calculations are performed using a plane wave basis set Vienna Ab-initio Simulation Package (VASP) code[32]. The generalized gradient approximation (GGA) is used including Perdew–Burke–Ernzerhof (PBE) functional to describe exchange and correlation, and the interaction between core and valence electrons is described with the projector augmented wave (PAW) method[33]. The energies and residual forces are converged to 10$^{-6}$ eV and 0.02 eV/Å, respectively. Both substrates and nanoparticles are accordingly simplified to facilitate the calculation according to our TEM result in Fig. 2. For MnO$_x$/Ni system, the plane wave cut-off energy is set to 400 eV for total energy calculation. The optimized crystal structure of Ni on a 6 × 6 × 6 k-point grid is cubic phase with a = 3.517 Å, which is in good agreement with experimental values[34]. The periodic slab model is used to simulate the Ni(111) surface with a four-layer supercell of p(3 × 3). The two bottom layers are fixed and other atoms are fully relaxed. The vacuum region is 15 Å. The MnO$_x$ segregation on Ni(111) surface is mimicked by a system containing MnO$_x$ clusters with 6 Mn atoms and 12, 10, or 9 O atoms. A 2 × 2 × 1 k-point grid is used for Brillouin zone sampling of the MnO$_x$/Ni(111) system. The different configurations of MnO$_x$ cluster on Ni(111) surface are considered, while the structures and adsorption energy are shown in Supplementary Fig. 8a after optimization. For M/CeO$_2$(111) system where the M is Cu, Ni or Ni–Cu, the cut-off energy is 460 eV. The DFT+U approach is used to describe the localized 4f electronic states in Ce, and we choose a Hubbard-U value of 5 eV for the Ce 4f states. The optimized crystal structure of CeO$_2$ on a 4 × 4 × 4 k-point grid is cubic phase with a = 5.413 Å, which is in good agreement with experimental values[35]. A p(2 × 2) superstructure with six-layer (96 atoms) with CeO$_2$(111) surface is selected. The three bottom layers are fixed with other atoms fully relaxed, and the vacuum region is 20 Å in thickness. The interface structure of four clusters (Ni$_9$, Ni$_{10-1}$, Ni$_{10-2}$, Ni$_{11}$) on CeO$_2$(111) surface are calculated. The Ni$_{11}$ cluster structure is selected on the CeO$_2$(111) surface in this work. Supplementary Fig. 8b shows the structure and binding energy after optimization. We replace Ni atoms with Cu atoms to obtain Cu/CeO$_2$(111) system. We then replace three Ni atoms with Cu atoms to obtain more uniform distribution alloy structure. Spin-polarized calculations are applied throughout all the calculations of M/CeO$_2$(111) systems. A 2 × 2 × 1 k-point grid is used for the Brillouin zone sampling of M/CeO$_2$(111) systems.

For Ni/TiO$_2$ system, the optimized lattice parameters of rutile TiO$_2$ are a = b = 4.646 and c = 2.966 Å with the 5 × 5 × 8 k-point grid, which is consistent with earlier reports[33]. A p(3 × 3) superstructure with three Ti–O layers (54 Ti and 108 O atoms) of the (101) surface of TiO$_2$ is used to simulate the periodic slab model. The bottom Ti–O layers are fixed and the top two Ti–O layers are fully relaxed. The vacuum region is 20 Å in thickness. We also study the energy values and adsorption energies of $CO_2$ with different cut-off energies, as shown in Supplementary Table 5, and we show that the adsorption energy of $CO_2$ generally remain unchanged though the energy values change with different cut-off energies. We then use the cut-off energy of 450 eV to calculate the adsorption energy of $CO_2$ as listed in Supplementary Fig. 9e and Supplementary Table 4. The structures of four clusters (Ni$_8$, Ni$_9$, Ni$_{10}$, Ni$_{11}$) on TiO$_2$(101) surface is constructed. The structure and binding energy of four clusters after optimization are shown in Supplementary Fig. 8c. The Ni$_{11}$ cluster structure is selected on the TiO$_2$(101) surface in this work. The Brillouin zone is sampled using a 3 × 3 × 1 k-point mesh. The adsorption energy of $CO_2$ is calculated using $E_{ads} = E_{total} - E_{CO_2} - E_{slab}$[36], where $E_{total}$ is the total energy of the adsorption system, $E_{CO_2}$ and $E_{slab}$ are the energy of the $CO_2$ in gas phase and the energy of the system without adsorption, respectively. Based on this definition, a more negative adsorption energy corresponds to a stronger adsorption. The adsorption energy ($E_a$) of MnO$_x$ is calculated as $E_a = E_{total} - E_{MnO_x} - E_{slab}$, where $E_{total}$ is the total energy of the adsorption system, $E_{MnO_x}$ and $E_{slab}$ are the energy of the MnO$_x$ and the energy of the system without adsorption, respectively. We calculate the binding energy ($E_b$) of the metal atoms on the surfaces using $E_b = 1/n \, (E_{tot} - E_{sur} - nE_{Ni})$, where $E_{tot}$ is the total energy of metal atoms on the surface, $E_{sur}$ is the energy of the surface, $E_{Ni}$ is the energy of a single Ni atom, and $n$ is the number of Ni atoms.

## Data availability

Data are available from the source file: https://yunpan.360.cn/surl_yFSD7aQKU9Z (Code:24d4)

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

## Acknowledgements

K.X. would like to acknowledge the funding support form Natural Science Foundation of China (91845202, 21750110433), Dalian National Laboratory for Clean Energy (DNL180404), and Strategic Priority Research Program of Chinese Academy of Sciences (XDB2000000). J.T.S.I. would like to acknowledge support from the EPSRC for the Emergent Nanomaterials Critical Mass project (EP/R023603/1).

## Author contribution

W.W. and L.G. conducted the experiments. W.W. and L.G. contributed equally to this work. J.P.L. and F.C. polished the English and provided the useful discussion. K.X. and J. T.S.I. supervised the work. All authors were involved in data analysis and discussion.

## Additional information

**Competing interests:** The authors declare no competing interests.

**Journal Peer Review Information:** *Nature Communications* thanks the anonymous reviewers for their contribution to the peer review of this work. Peer reviewer reports are available.

