## [Peer Review File · Nature Communications]

Reviewers' comments:

Reviewer #1 (Remarks to the Author):

The authors have revised the manuscript according to the comments. This manuscript is suggested to be accepted for publication.

Reviewer #2 (Remarks to the Author):

Authors report the highly active electrode based on exsolution approach. Results are interesting, and of impact. However, the following points have made the reviewer reluctant to recommend the article for publication in Nature Communication.

1. Authors insist a synergistic control of materials functions and interface architectures, however, the reviewer fails to understand clearly the meaning. What authors have conducted is to synthesize materials that will lead to exsolution of some species. The reviewer did not understand what and how materials functions are controlled, but did not control interface architectures, nor how the interface architectures are controlled. For example, how to control the facets of exsolved phase and the matrix surface, or sized/density of exsolved phase? To the reviewer's eye, it can be mentioned as effective exsolution approach, but cannot be generalized as "interface", nor "engineering". Therefore, more specific journals focusing on energy materials, catalysis, electrochemistry are recommended.

2. Related to the above-mentioned comment, the interface structure cannot be well-understood. Authors calculated the vibrational frequency of CO₂ assuming stable surface facet of oxide. If authors insist that the obtained results agree well with experimental observation, it clearly means that experimentally observed spectra correspond to the most stable bulky surface of oxide, and do not represent the spectra of CO₂ at the "interface" between oxide and metal. However in reality, the experimental and theoretical results match only in the case shown in Fig 4d. Other cases show big deviations.

In addition, careful discussion of interface structure shown in Fig. 2 and 5 is missing. Crystalline phase is obvious in Fig. 2, while amorphized structure can be seen in Fig. 5. One will notice a big difference, rather than be convinced that the model structures used in theoretical calculations represent the essential part of real structures.

3. Authors descriptions are not ready for the submission to reputed journal.

Page 5 line 204 supplementary Figure 7a does not show the possible CO₂ adsorption configuration

Page 9 line 345 and 357 Figure S10 does not show the structures mentioned in the text

Supplementary Information Figure 7 the authors do not show MnO₂ on Ni(111) models nor the adsorption energies

The authors are omitting the information regarding TiO₂ in Tables 1 and 2.

Page 1 line 16 SOFC was not defined

Page 2 line 45 YSZ was not defined

Page 2 line 78 typo "across"

Page 3 line 102 XRD was not defined

Page 3 line 107 XPS was not defined

Page 3 line 111 TGA was not defined, it was defined on line 120

Page 3 line 119 the metal are exsolved, change to the metal is exsolved

Page 4 line 132 In Figure 1e, In is bold

Page 4 line 149 HR-TEM was not defined

Page 5 line 182 FT-IR was not defined

Page 5 from line 194 to the end of the document (including figures and supplementary information) remove the space between the material and Miller index:

CeO₂ (111) change to CeO₂(111)

TiO₂ (110) change to TiO₂(110)

MnO₂ (111) change to MnO₂(111)

Page 7 line 266 typo creia, change it to ceria

Page 8 lines 325-326 generalized gradient approximation instead of generalised gradient approach

Page 9 line 349 calculation have been applied throughout. Throughout what?

Page 9 line 350 Cu、 change to Cu,

Figure 4a,c,e change CO 2 to CO₂

Supplementary information Page 1 line 1 and 2 have different format, NiO/15%NiMn₂O₄... Ni/11% MnO_x

4. For the reproducibility of the DFT calculations, provide detailed information on the total layer of TiO₂. In addition, the influence of different cut off energy should be mentioned because the different cut off will give us different energy values, thus one will not be able to compare the results each other as obtained.

Reviewers' comments:

Reviewer #1 (Remarks to the Author):

The authors have revised the manuscript according to the comments. This manuscript is suggested to be accepted for publication.

Answer: Thank you very much for your suggestion.

Reviewer #2 (Remarks to the Author):

Authors report the highly active electrode based on exsolution approach. Results are interesting, and of impact. However, the following points have made the reviewer reluctant to recommend the article for publication in Nature Communication.

1. Authors insist a synergistic control of materials functions and interface architectures; however, the reviewer fails to understand clearly the meaning. What authors have conducted is to synthesize materials that will lead to exsolution of some species. The reviewer did not understand what and how materials functions are controlled, but did not control interface architectures, nor how the interface architectures are controlled. For example, how to control the facets of exsolved phase and the matrix surface, or sized/density of exsolved phase? To the reviewer's eye, it can be mentioned as effective exsolution approach, but cannot be generalized as "interface", nor "engineering". Therefore, more specific journals focusing on energy materials, catalysis, electrochemistry are recommended.

Answer: We do not fully agree with the referee on this broad use of the term exsolution. Exsolution is simply the decomposition of one phase into two phases and can vary significantly in its nature. We could easily have used this as a generic term to satisfy these comments, but this would have lost the important subtleties of the exsolution and other processes occurring across these different systems. We use chemical control to engineer the interface to create nanoparticles via exsolution, but also modify the surface, and create critical vacancy sites so it is much more than even a careful exsolution process. We intended to suggest that we could use chemical control to engineer the interface, but are willing to accept the referee's views that these terms might suggest a physical, external modification of the interface. We therefore suggest a change in title to "Enhanced CO₂ electrolysis at redox manipulated interfaces".

2. Related to the above-mentioned comment, the interface structure cannot be well-understood. Authors calculated the vibrational frequency of CO₂ assuming stable surface facet of oxide. If authors insist that the obtained results agree well with experimental observation, it clearly means that experimentally observed spectra correspond to the most stable bulky surface of oxide, and do not represent the spectra of CO₂ at the "interface" between oxide and metal. However in reality, the experimental and theoretical results match only in the case shown in Fig 4d. Other cases show big deviations. In addition, careful discussion of interface structure shown in Fig. 2 and 5 is missing. Crystalline phase is obvious in Fig. 2, while amorphized structure can be seen in Fig. 5. One will notice a big difference, rather than be convinced that the model structures used in theoretical calculations represent the essential part of real structures.

Answer: Thank you very much. We agree with your comments, that the vibrational frequency is actually just a reflection of CO₂ adsorption on oxide surface, it was not our intention to suggest that the adsorption was specifically at the actual metal-oxide interfaces. This has been rectified. We also re-calculate the vibrational frequency of CO₂ assuming a stable surface facet of oxide to simplify the calculation, though the oxide surface is extremely complicated and is composed of poly-crystalline states. We add careful discussion of interface structure in Figure 2 and 5. Both substrates and nanoparticles are crystalline phases in **Figure 2**, and then we construct the

cluster model also in crystalline phase on substrate according to TEM results. After structure optimization, the proposed clusters have been transformed into amorphized structure, which could be due to that the cluster is not large enough to maintain the crystalline state in such a small scale. As shown in **Supplementary Table 1**, the calculated vibrational frequencies match the experimental data while the adsorption configurations are shown in **Supplementary Figure 7**. The oxide surface not the metal-oxide interface would dominate the vibrational signal for CO₂ adsorption. And the presence of oxygen vacancy linked to dopant would further facilitate the CO₂ reduction on oxide surfaces in an exothermic process.¹

Supplementary Table 1. Vibrational frequencies in cm⁻¹ of CO₂ species and CO₃²⁻ species adsorbed on MnO, CeO₂ and TiO₂ surfaces.

system	species	parameter	Figure	C-O (Å)	ν (cm ⁻¹)
MnO	CO ₂	experimental	-	-	2361
		computational	7a	1.17	2372
	CO ₃ ²⁻	experimental	-	-	1382
		computational	7b	1.31	1364
CeO ₂	CO ₂	experimental	-	-	2360
		computational	7c	1.17	2372
	CO ₃ ²⁻	experimental	-	-	1389
		computational	7d	1.26	1390
TiO ₂	CO ₂	experimental	-	-	2359
		computational	7e	1.18	2355
	CO ₃ ²⁻	experimental	-	-	1397
		computational	7f	1.29	1398

Supplementary Figure 7. Structures of CO₂ species and CO₃²⁻ species adsorbed on MnO, CeO₂ and TiO₂ surfaces. Cerium in apricot cream, titanium in silvery white, manganese in purple, carbon in grey, oxygen in red and oxygen of CO₂ species and CO₃²⁻ species in yellow for clear.

3. Authors descriptions are not ready for the submission to reputed journal. Page 5 line 204 supplementary Figure 7a does not show the possible CO₂ adsorption configuration. Page 9 line 345 and 357 Figure S10 does not show the structures mentioned in the text. Supplementary Information Figure 7 the authors do not show MnO₂ on Ni(111) models nor the adsorption energies. The authors are omitting the information regarding TiO₂ in Tables 1 and 2.

Answer: Thank you for your comments. We further correct the adsorption configurations of CO₂ on our models and then perform additional calculation of CO₂ adsorption as shown in **Supplementary Figure 9**. We show that the CO₂ adsorption is favorable at metal surface, oxide surface and metal-oxide interface. The exsolution of nanoparticles would enhance CO₂ adsorption and activation as shown in **Figure 5**. The MnO_x on Ni(111) models

have been added in **Supplementary Figure 8a**. We also add the calculation details of TiO_2 in **Supplementary Table 4**.

Supplementary Figure 9 The adsorption configurations of CO_2 on $\text{MnO}_x/\text{Ni}(111)$, $\text{M}/\text{CeO}_2(111)$ and $\text{Ni}/\text{TiO}_2(101)$ systems, respectively. (a1-a2) $\text{Mn}_6\text{O}_{12}/\text{Ni}(111)$, (a3) $\text{Mn}_6\text{O}_{10}/\text{Ni}(111)$ and (a4) $\text{Mn}_6\text{O}_9/\text{Ni}(111)$ systems; (b-d) $\text{M}/\text{CeO}_2(111)$ systems, M represents Cu, Ni and Ni-Cu clusters respectively; (e1-e4) $\text{Ni}/\text{TiO}_2(101)$ systems. Nickel in blue, copper in orange, cerium in apricot cream, titanium in silvery white, manganese in purple, carbon in grey and oxygen in red.

Figure 5 | Different adsorption configurations of CO_2 on the system. (a-b), The MnO_x/Ni system (c-d), the Ni-Cu/ CeO_2 system. (e-f), the Ni/TiO_2 system. (top: clean surfaces, bottom: defective surfaces, 1 and 4: side views, 2 and 5: top views, 3 and 6: contour plots of electronic charge density difference). Nickel in blue, copper in orange, cerium in silvery white, titanium in pale, manganese in purple, oxygen in red and carbon in grey.

Figure 8 Adsorption energies of MnO_x on Ni(111) surface and binding energies of different clusters on $CeO_2(111)$ surface and $TiO_2(101)$ surface. (a1-a2) different Mn_6O_{12} shapes while (a3-a4) different Mn_6O_9 shapes on the Ni(111) surface. (b1-b4) different Ni clusters shapes on the $CeO_2(111)$ surface. (c1-c4) different Ni clusters shapes on the $TiO_2(101)$ surface. Nickel in blue, cerium in apricot cream, titanium in silvery white, manganese in purple and oxygen in red.

Supplementary Table 4. Geometrical parameters and calculated adsorption energies of CO_2 species on Ni/ $TiO_2(101)$ systems. TiO_2 -I to TiO_2 -IV are the adsorption configurations of Figure 9 e1 to e4, TiO_{2-x} -I and TiO_{2-x} -II are the adsorption configurations of Figure 5 f1 and f4.

parameter	CO_2	TiO_2 -I	TiO_2 -II	TiO_2 -III	TiO_2 -IV	TiO_{2-x} -I	TiO_{2-x} -II
C-Ni (Å)	-	1.93	2.06	1.96	-	1.87	1.78/1.87
O1-Ni (Å)	-	1.97	1.97	1.95	-	2.01	-
O2-Ni (Å)	-	1.97	-	1.95	-	1.99	-
O-Ti (Å)	-	-	2.15	-	2.04/1.99	-	1.67
C-O1 (Å)	1.18	1.28	1.28	1.29	1.20	1.26	1.22
C-O2 (Å)	1.18	1.27	1.28	1.30	1.28	1.29	3.85
O-C-O (°)	180	130.8	126.0	127.2	134.3	131.2	106.8
E_{ads} (eV)	-	-1.39	-1.27	-1.44	-0.25	-1.16	-1.94

Page 1 line 16 SOFC was not defined. Page 2 line 45 YSZ was not defined. Page 2 line 78 typo “acoss”. Page 3 line 102 XRD was not defined. Page 3 line 107 XPS was not defined. Page 3 line 111 TGA was not defined, it was defined on line 120. Page 3 line 119 the metal are exsolved, change to the metal is exsolved. Page 4 line 132 In Figure 1e, In is bold. Page 4 line 149 HR-TEM was not defined. Page 5 line 182 FT-IR was not defined. Page 5 from line 194 to the end of the document (including figures and supplementary information) remove the space between the material and Miller index: $CeO_2(111)$ change to $CeO_2(111)$, $TiO_2(110)$ change to $TiO_2(110)$, $MnO_2(111)$ change to $MnO_2(111)$, Page 7 line 266 typo crea, change it to ceria, Page 8 lines 325-326 generalized gradient approximation instead of generalised gradient approach, Page 9 line 349 calculation have been applied throughout. Throughout what? Page 9 line 350 Cu change to Cu, Figure 4a,c,e change CO_2 to CO_2 , Supplementary information Page 1 line 1 and 2 have different format, NiO/15%NiMn₂O₄... Ni/11% MnO_x

Answer: Yes, thanks very much. We have defined “SOFC”, “YSZ”, “XRD”, “XPS”, “TGA”, “HR-TEM” and “FT-IR” in relevant positions and corrected the expressions as you mentioned above.

For the reproducibility of the DFT calculations, provide detailed information on the total layer of TiO_2 . In

addition, the influence of different cut off energy should be mentioned because the different cut off will give us different energy values, thus one will not be able to compare the results each other as obtained.

Answer: Yes, thanks very much. We further revise the detail description for the total layer of TiO₂ in revision and re-calculate the energy values with different cut off energy to explore the corresponding influence. For Ni/TiO₂ system, we use the optimized crystal structure and lattice parameters of rutile TiO₂ on a 5×5×8 k-point grid, with a=b=4.646, c=2.966 Å. A p(3×3) superstructure with three Ti-O layers (54 Ti and 108 O atoms) of the (101) surface of TiO₂ is used to simulate the periodic slab model. The bottom Ti-O layer is fixed and the top two Ti-O layers are fully relaxed. The vacuum region is 20 Å in thickness.

We re-calculate the adsorption energy of CO₂ with different cut off energy from 300 to 650 eV as shown in Supplementary Table 5. Although the different cut off energies will give us different energy values, the adsorption energy of CO₂ adsorption remain generally unchanged as calculated by $E_{ads} = E_{total} - E_{CO_2} - E_{slab}$, where E_{total} is the total energy of the adsorption system, E_{CO_2} and E_{slab} are the energy of the CO₂ in gas phase, and the energy of the system without adsorption, respectively. In our work, the cut-off energy is set to 450 eV for TiO₂ system in our calculation, which is well consistent with previous reports^{2,3}.

Table 5 The energy values and adsorption energy of CO₂ with different cut-off energy. TiO₂-I to TiO₂-IV are the adsorption configurations of Figure 9 e1 to e4.

Cut-off (eV)	300	350	400	450	500	550	600	650
E_{CO_2} (eV)	-23.16	-23.02	-23.98	-22.95	-22.94	-22.95	-22.96	-22.97
E_{TiO_2} (eV)	-1309.35	-1300.10	-1296.86	-1296.48	-1296.18	-1296.19	-1296.47	-1296.77
E_{TiO_2-I} (eV)	-1333.91	-1324.54	-1322.24	-1320.82	-1320.45	-1320.53	-1320.82	-1321.13
E_{TiO_2-II} (eV)	-1333.79	-1324.02	-1322.12	-1320.70	-1320.19	-1320.40	-1320.63	-1321.01
E_{TiO_2-III} (eV)	-1333.97	-1324.60	-1322.28	-1320.87	-1320.50	-1320.58	-1320.87	-1321.15
E_{TiO_2-IV} (eV)	-1332.82	-1323.41	-1321.09	-1319.68	-1319.30	-1319.38	-1319.68	-1319.98
E_{ads-I} (eV)	-1.41	-1.42	-1.40	-1.39	-1.33	-1.39	-1.39	-1.39
E_{ads-II} (eV)	-1.29	-0.90	-1.29	-1.27	-1.07	-1.26	-1.20	-1.27
$E_{ads-III}$ (eV)	-1.47	-1.47	-1.45	-1.44	-1.37	-1.43	-1.44	-1.41
E_{ads-IV} (eV)	-0.32	-0.29	-0.26	-0.25	-0.18	-0.24	-0.25	-0.24

Reference

1. Acharya, D.P., Camillone, N., Sutter, P., CO₂ Adsorption, Diffusion, and Electron-Induced Chemistry on Rutile TiO₂(110): A Low-Temperature Scanning Tunneling Microscopy Study, *Journal of Physical Chemistry C* 115, 12095-12105 (2011).
2. Wang, D., Sheng, T., Chen, J., Wang, H.F. & Hu, P. Identifying the key obstacle in photocatalytic oxygen evolution on rutile TiO₂. *Nature Catalysis* 1, 291-299 (2018).
3. Wang, D., Liu, Z.P. & Yang, W.M. Revealing the Size Effect of Platinum Cocatalyst for Photocatalytic Hydrogen Evolution on TiO₂ Support: A DFT Study. *ACS Catalysis* 8, 7270-7278 (2018).

REVIEWERS' COMMENTS:

Reviewer #2 (Remarks to the Author):

The reviewer found significant improvements in the revised manuscript.
The manuscript is recommended for the publication after modifying the following point.

The authors describe "The growth of exsolved interfaces on porous electrode scaffolds would create "three phase boundary (TPB)" with the convergence of electronic conduction phase, ionic conduction phase and gaseous phase."

Because the exsolved phases take the islandic morphology, it is misleading to use the term "conduction".

Unlike the typical TPB in solid oxide cell, this TPB is not for the electrochemical reaction.

Metal island on oxide may act as an electron reservoir, metal will act as catalyst facilitating the oxygen exchange at the TPB while the electron does not conduct through the metal network which is the process in typical solid oxide cell.

Oxide island on metal may act as an oxygen reservoir, but exsolved oxide phase does not play any role for providing oxide ion conduction pathway.

To clearly differentiate the TPB of exsolved island and typical TPB in solid oxide cell, authors are encouraged to use more careful phrasings.

RESPONSE TO REVIEWER

REVIEWERS' COMMENTS:

Reviewer #2 (Remarks to the Author):

The reviewer found significant improvements in the revised manuscript. The manuscript is recommended for the publication after modifying the following point.

The authors describe "The growth of exsolved interfaces on porous electrode scaffolds would create "three phase boundary (TPB)" with the convergence of electronic conduction phase, ionic conduction phase and gaseous phase." Because the exsolved phases take the islandic morphology, it is misleading to use the term "conduction". Unlike the typical TPB in solid oxide cell, this TPB is not for the electrochemical reaction. Metal island on oxide may act as an electron reservoir, metal will act as catalyst facilitating the oxygen exchange at the TPB while the electron does not conduct through the metal network which is the process in typical solid oxide cell. Oxide island on metal may act as an oxygen reservoir, but exsolved oxide phase does not play any role for providing oxide ion conduction pathway. To clearly differentiate the TPB of exsolved island and typical TPB in solid oxide cell, authors are encouraged to use more careful phrasings.

Answer: Thank you very much for your comments. Yes, we totally agree with that the exsolved interfaces are different from the typical three phase boundary in solid oxide cells. The active interface for metallic cathodes where CO₂ splitting reactions proceed is normally located on the electrolyte surface with intimate contact with porous metal phase. Only the exsolved metal-oxide interfaces at the interfaces between electrolyte surface and porous metal phase can function well for CO₂ electrolysis. Similar to metal cathodes, CO₂ electrolysis in the traditional Ni-YSZ composite cathode proceeds at interfaces with the convergence of electronic conduction phase, ionic conduction phase and gaseous phase. Exsolved interfaces at these three phase boundaries would be effective to enhance CO₂ electrolysis. However, a ceramic cathode with mixed conduction has a more extensive surface, functioning as electrochemical reaction region. Exsolved metal-oxide interfaces on the whole surface would be possible to enhance the surface electrochemical reactions. We have carefully revised this part in revision.